# Intracranial Hemorrhage in a Patient with TAFRO Syndrome Treated with Cyclosporine A and Rituximab

**DOI:** 10.3390/medicina57090971

**Published:** 2021-09-16

**Authors:** Yuina Akagi, Takashi Kato, Yusuke Yamashita, Hiroki Hosoi, Shogo Murata, Shuto Yamamoto, Kenji Warigaya, Taisei Nakao, Shinichi Murata, Takashi Sonoki, Shinobu Tamura

**Affiliations:** 1Department of Hematology/Oncology, Wakayama Medical University, Wakayama 6418509, Japan; kurenainoyuni@yahoo.co.jp (Y.A.); ktakashi@wakayama-med.ac.jp (T.K.); yyyamash@wakayama-med.ac.jp (Y.Y.); hiro-hosoi@hotmail.co.jp (H.H.); shogo@wakayama-med.ac.jp (S.M.); sonoki@wakayama-med.ac.jp (T.S.); 2Department of Internal Medicine, Naga Municipal Hospital, Wakayama 6496414, Japan; t.nakao@nagahp.jp; 3Department of Nephrology, Wakayama Medical University, Wakayama 6418509, Japan; shu-yama@wakayama-med.ac.jp; 4Department of Diagnostic Pathology, Wakayama Medical University, Wakayama 6418509, Japan; ketunai@wakayama-med.ac.jp (K.W.); smurata@wakayama-med.ac.jp (S.M.)

**Keywords:** TAFRO syndrome, thrombotic microangiopathy, cyclosporine A, rituximab, intracranial hemorrhage

## Abstract

TAFRO syndrome, a rare subtype of idiopathic multicentric Castleman disease, manifests as thrombocytopenia, anasarca, fever, reticulin fibrosis, and organomegaly. Thrombotic microangiopathy, including renal dysfunction, is frequently associated with this syndrome. TAFRO syndrome can be life threatening and show rapid progression, and the diagnosis and management of this disorder remain challenging. A 48-year-old woman was diagnosed with TAFRO syndrome complicated by thrombotic microangiopathy based on the clinical and histopathological findings. After receiving high-dose steroids, her thrombocytopenia and anasarca did not improve. The patient subsequently received a combination of cyclosporine A and rituximab as second-line therapy, which resulted in a significant gradual improvement in the clinical symptoms. Meanwhile, her platelet count increased to more than 40 × 10^9^/L; however, she developed intracranial hemorrhage. Following surgical evacuation, the patient recovered with an achievement of sustained remission. Based on these findings, attention should be paid to life-threatening bleeding associated with local thrombotic microangiopathy even when intensive treatment is administered for TAFRO syndrome.

## 1. Introduction

TAFRO syndrome is a rare systemic inflammatory disorder characterized by thrombocytopenia, anasarca, reticulin fibrosis of the bone marrow, renal dysfunction, and organomegaly [1,2]. In 2010, three patients with TAFRO syndrome were first described in Japan [3]. Histopathological evaluation of the lymph nodes from patients with TAFRO syndrome revealed characteristics that were similar to those in patients with idiopathic multicentric Castleman’s disease, particularly the hyaline-vascular variant [1,2]. Nonetheless, TAFRO syndrome clinically differs from typical idiopathic multicentric Castleman’s disease, considering that the former is usually a rapid, aggressive, and life-threatening condition. However, TAFRO syndrome was later classified as a variant of idiopathic multicentric Castleman’s disease based on the histopathological similarities [4]. Notably, studies have found that corticosteroids have limited efficacy against TAFRO syndrome, which generally requires additional immunosuppressants, including cyclosporine A, tacrolimus, and cyclophosphamide [2,5,6,7]. Considering that the pathogenesis and symptomatology of TAFRO syndrome have been associated with the excessive release of interleukin-6 (IL-6) and vascular endothelial growth factor (VEGF) [1,2], tocilizumab, a humanized anti-IL-6 receptor antibody, has emerged as a therapeutic option for patients with TAFRO syndrome. Moreover, rituximab, an anti-CD20 antibody, has also been successfully used in combination with corticosteroids and/or immunosuppressants [3,8,9,10]. In 2016, Masaki et al. proposed diagnostic criteria, severity classification, and treatment strategy for TAFRO syndrome [2]. Since then, case reports of TAFRO syndrome have gradually increased worldwide, although its etiology remains unclear.

Herein, we present a case of TAFRO syndrome diagnosed using bone marrow and kidney histopathology that suddenly developed intracranial hemorrhage during combination therapy with cyclosporine A and rituximab.

## 2. Case Report

A 48-year-old woman was admitted because of persistent fever and abdominal distention for 2 weeks. There was no notable previous medical or familiar history, and the patient had never smoked or consumed alcohol. Within a week after admission, she developed pleural effusion, ascites, hepatosplenomegaly, renal dysfunction, and thrombocytopenia of unknown etiology. Despite antibiotic and diuretic administration, her general condition rapidly worsened. Given the suspicion of a hematological disorder, the patient was transferred to our hospital for further examination.

Upon transfer to our hospital, she was found to have a fever of 37.3 °C, heart rate of 90 beats/min, blood pressure of 144/83 mmHg, and oxygen saturation of 97% on room air. Physical examination revealed diffusely decreased respiratory sounds in the right lung, severe abdominal distension, and pitting edema of the lower extremities. No peripheral lymph nodes were palpable, while laboratory studies revealed mild thrombocytopenia (platelet count of 123 × 10^9^/L) and elevated levels of alkaline phosphatase (325 IU/L), soluble interleukin-2 receptor (981 U/mL), C-reactive protein (CRP) (3.0 mg/dL), and creatinine (1.71 mg/dL), along with proteinuria and microscopic hematuria (Table 1). Although the patient was positive for anti-Sjögren’s-syndrome-related antigen A and anti-histidyl-tRNA synthetase antibodies, she did not fulfill the criteria for connective tissue disease. Screening tests for hepatitis B virus, hepatitis C virus, cytomegalovirus, and human immunodeficiency virus were negative. Although her serum IL-6 level was almost normal (5.0 pg/mL; reference range < 4.0 pg/mL), elevated serum VEGF levels were noted (256 pg/mL; reference range < 38.3 pg/mL). Computed tomography (CT) revealed right pleural effusion, massive ascites, and hepatosplenomegaly without enlarged lymph nodes (Figure 1), and bone marrow biopsy showed hypercellular marrow with megakaryocyte hyperplasia (Figure 2, arrowheads) and mild reticulin fibrosis (Figure 2, arrow).

On the third day of admission, renal biopsy was performed to evaluate acute kidney injury regardless of the decrease in platelet count (79 × 10^9^/L). The aforementioned clinical findings fulfilled all three major and three minor criteria for TAFRO syndrome according to the Japanese diagnostic criteria published in 2015 [2]. On the fifth day of admission, a diagnosis of TAFRO syndrome was established based on the clinical, laboratory, and histopathological findings. Additionally, the patient was considered to have moderately severe disease (score of five out of 12 points) [2]. Moreover, the final report detailing the pathological findings of the renal biopsy revealed prominent swelling and increased number of glomerular endothelial cells with glomerular endothelial injury due to renal thrombotic microangiopathy (Figure 3). These renal biopsy findings were also consistent with those of TAFRO syndrome.

The clinical course after admission to our hospital is shown in Figure 4. Soon after the diagnosis of TAFRO syndrome, the patient was initially treated with 60 mg (1 mg/kg of body weight) of oral prednisolone (PSL) daily; however, an adequate response was not achieved. Therefore, methylprednisolone pulse therapy (1000 mg/day) was administered for three consecutive days followed by PSL 60 mg (1 mg/kg of body weight) per day. Considering that her symptoms and clinical course did not respond to steroid therapy, platelet transfusion and ascites paracentesis were performed. Additionally, she often experienced a sudden and transient drop in blood pressure, pulse, and oxygen saturation. On the 11th day of admission, continuous hemodiafiltration was required because of oliguric renal failure. Subsequently, oral cyclosporine A (CyA, 4 mg/kg/day in two divided doses) and weekly rituximab (375 mg/m^2^, four infusions) were selected as the second-line treatment, along with steroid tapering (Figure 4). Although the patient’s serum IL-6 level was found to be within the normal range immediately before the first infusion of rituximab, her serum VEGF level was still high at 92 pg/mL (reference range < 38.3 pg/mL). After initiating second-line treatment, renal dysfunction, thrombocytopenia, and CRP levels gradually improved, and hemodialysis was terminated on the 38th day of admission (Figure 4). The patient displayed a stable circulatory status, with a gradual decrease in anasarca and ascites.

On the 39th day of admission, the patient developed severe headache and vomiting. A cranial CT scan revealed cerebral and subarachnoid hemorrhage (Figure 4), despite having a platelet count of 42 × 10^9^/L with normal coagulation parameters. The patient subsequently underwent an emergency craniotomy for evacuation. Postoperative angiography revealed no cerebral aneurysm or brain arteriovenous malformation. Although prolonged rehabilitation was required after the surgery, the patient exhibited no residual neurological sequelae. On the 84th day of admission, the patient was discharged ambulatory and maintained remission and normal renal function with oral CyA (150 mg/day, 3 mg/kg/day in two divided doses) monotherapy for more than a year.

## 3. Discussion

An increasing number of case reports with TAFRO syndrome has recently described renal percutaneous biopsy among the diagnostic procedures, despite low platelet counts. Given that our patient had a platelet count exceeding 7 × 10^9^/L, renal biopsy was performed without any complications. The main pathological findings of renal involvement in TAFRO syndrome are thrombotic microangiopathy-like glomerulopathy and membranoproliferative glomerulonephritis-like lesions [11,12]. In particular, thrombotic microangiopathy-like glomerulopathy is considered a typical pathological characteristic of acute-phase TAFRO syndrome [11,12]. According to the pathologist, the renal histology in our patient was consistent with thrombotic microangiopathy-like findings, suggesting that our patient was in the acute phase of TAFRO syndrome. Moreover, high levels of serum VEGF have been observed in most patients with TAFRO syndrome [1,2]. The highly elevated serum VEGF levels in our patient upon diagnosis could have aggravated endothelial glomerular injury, which was followed by local thrombotic microangiopathy.

Only one case of intracranial hemorrhage during the clinical course of TAFRO syndrome has been previously reported. This particular case involved an elderly patient with TAFRO syndrome who received steroids and tocilizumab and developed cerebral hemorrhage despite a platelet count of 80 × 10^9^/L, with the occurrence of death soon after due to aspiration pneumonia [13]. To our knowledge, this is the first report of a patient surviving TAFRO syndrome complicated by an intracranial hemorrhage, although several cases of intracranial hemorrhage during the course of idiopathic multicentric Castleman’s disease have been published [14,15]. The risk of bleeding in TAFRO syndrome has been attributed to thrombocytopenia. However, although our patient had achieved a platelet count of 42 × 10^9^/L when intracranial hemorrhage occurred, no cerebral aneurysm or arteriovenous malformation on cerebral angiography were observed. As such, the findings presented herein, as well as in a previously reported case [13], suggest little correlation between the platelet count and intracranial hemorrhage in patients with TAFRO syndrome. Moreover, we assumed that our patient developed endothelial brain injury due to local thrombotic microangiopathy, similar to that observed in the kidneys, which resulted in an intracranial hemorrhage. Therefore, although intensive immunosuppressive therapy is required for the clinical management of TAFRO syndrome, attention should also be paid to life-threatening bleeding.

Although the etiology of TAFRO syndrome, as well as the pathogenesis of idiopathic multicentric Castleman’s disease, are yet to be elucidated, the excessive production of proinflammatory cytokines, such as IL-6, can cause some of the symptoms associated with these diseases [9]. In fact, patients with TAFRO syndrome have also been shown to exhibit elevated serum levels of both IL-6 and VEGF [1,2,16]. Tocilizumab, an anti-IL-6 receptor antibody, has been widely used as a second-line treatment option for corticosteroid-resistant TAFRO syndrome [5,17,18,19,20]. Meanwhile, previous studies have also reported that patients with TAFRO syndrome who had normal serum IL-6 levels exhibited an inadequate response to tocilizumab [5,10,18,19,20]. In the present case, laboratory data before second-line treatment showed increased VEGF levels but normal IL-6 levels. Therefore, tocilizumab was not selected for our patient who was refractory to corticosteroid therapy. Some patients with tocilizumab resistance have been successfully treated with rituximab therapy [17,18,19,20], although the mechanism of action remains unclear. Recently, a retrospective study in Japan suggested that rituximab was a more effective second-line treatment than CyA or tocilizumab [20]. The clinical response to the additional rituximab in our patient also supports this notion, considering that the suppression of mature B cells by anti-CD20 antibodies improves clinical remission in patients with TAFRO syndrome.

Serum VEGF and circulating CD8+ T cells have been reported to be increased in non-Japanese patients with TAFRO syndrome [21,22,23,24]. Recent studies found that the mTOR (mammalian target of rapamycin) signaling pathway, which was critical to VEGF expression and CD8+ T cell activation, was elevated in such patients [22,23]. Furthermore, transcriptome analysis revealed that signature genes associated with interferon type I (IFN-I), which is one of the upstream regulators of mTOR signaling, were identified in CD8+ T cells, NK cells, and monocytes obtained from patients with TAFRO syndrome [24]. The overproduction of IFN-I in immune cells has been associated with the development of autoimmune diseases such as systemic lupus erythematous [25]. Notably, treatment with mTOR inhibitor has been shown to be effective in some patients with tocilizumab-refractory TAFRO syndrome [22]. These findings suggest that IFN-I-induced activation of mTOR signaling pathway serves a central role in the pathogenesis of TAFRO syndrome and develops autoimmune features such as thrombocytopenia.

## 4. Conclusions

Here, we have described our experience with a patient with TAFRO syndrome who was successfully treated with CyA and rituximab; however, she developed unexpected intracranial hemorrhage during the clinical course. Nonetheless, the efficacy of rituximab therapy for TAFRO syndrome requires further evaluation in prospective studies. Furthermore, clinicians should pay greater attention to life-threatening bleeding triggered by local thrombotic microangiopathy despite apparent improvements in the disease condition.

## Figures and Tables

**Figure 1 medicina-57-00971-f001:**
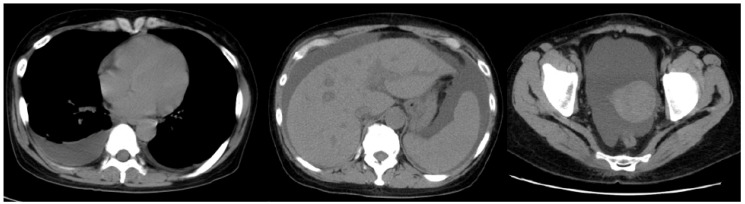
Chest and abdominal computed tomography (CT) images of the patient on transfer to our hospital. CT images show right-sided pleural effusion, ascites, and hepato-splenomegaly.

**Figure 2 medicina-57-00971-f002:**
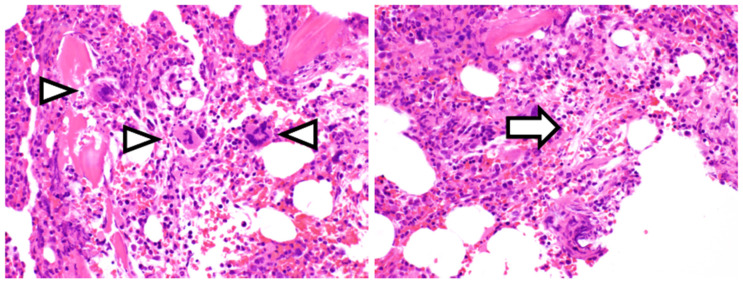
Bone marrow lesions in our patient. Hematoxylin and eosin staining shows a hypercellular marrow with megakaryocyte clusters (arrowheads) and mild reticulin fibrosis (arrow) (×200).

**Figure 3 medicina-57-00971-f003:**
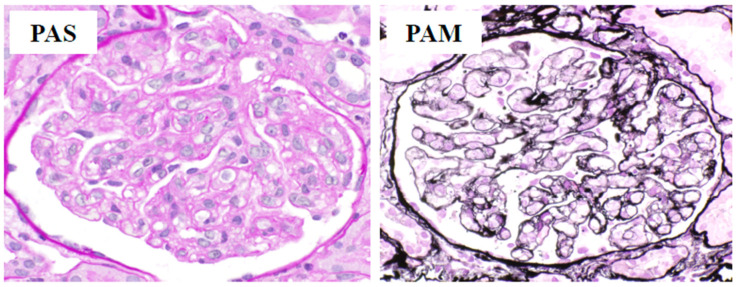
Kidney lesions in our patient. Periodic acid–Schiff and periodic acid methenamine silver staining shows prominent swelling and increased glomerular endothelial cells (×400).

**Figure 4 medicina-57-00971-f004:**
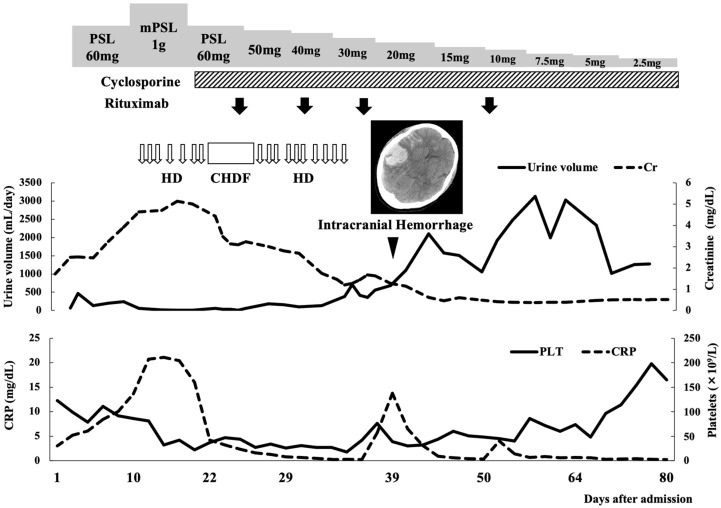
Clinical course of the patient after transfer to our hospital. Cranial computed tomography imaging of the patient shows a life-threating hemorrhage in the right subarachnoid space and cerebral hemisphere. CHDF, continuous hemodiafiltration; Cr, creatinine; CRP, C-reactive protein; HD, hemodialysis; PLT, platelets; PSL, prednisolone.

**Table 1 medicina-57-00971-t001:** Laboratory data of our patient with TAFRO syndrome at the transfer to our hospital.

Complete Blood Count			Chemistry			Calcium	9.4	mg/dL
White Blood Cells	11.1	×10^9^/L	Total Protein	5.9	g/dL	Total Bilirubin	0.5	mg/dL
Neutrophil	78	%	Albumin	3.0	g/dL	C-Reactive Protein	3.0	mg/dL
Eosinophil	0	%	Creatine Kinase	87	IU/L	IgG	1008	mg/dL
Basophil	0	%	Aspartate Transaminase	16	IU/L	IgA	60	mg/dL
Monocyte	7	%	Alanine Transaminase	7	IU/L	IgM	112	mg/dL
Lymphocyte	12	%	Lactate Dehydrogenase	187	IU/L	IgG4	11	mg/dL
Myelocytes	1	%	Alkaline Phosphatase	325	IU/L	TSH	2.38	µIU/mL
Metamyelocytes	2	%	γ-Glutamyl Transpeptidase	32	IU/L	Free T4	1.39	ng/mL
Red Blood Cells	4.14	×10^12^/L	Cholinesterase	285	IU/L	Soluble IL-2R	981	U/mL
Hemoglobin	10.7	g/dL	Amylase	63	IU/L	Ferritin	118	ng/mL
Reticulocytes	77	×10^9^/L	Creatinine	1.71	mg/dL			
Platelets	123	×10^9^/L	Uric Acid	8.4	mg/dL	**Coagulation system**		
			Blood Urea Nitrogen	32.7	mg/dL	APTT	32.3	s
**Serum Cytokine**			Sodium	139	mEq/L	Prothrombin time	1.04	s
IL-6 *	5.0	pg/mL	Potassium	4.3	mEq/L	Fibrinogen	423	mg/dL
VEGF *	256	pg/mL	Chloride	104	mEq/L	D-dimer	8.1	μg/mL

APTT, activated partial thromboplastin time; Ig, immunoglobulin; IL-2R, interleukin-2 receptor; IL-6, interleukin-6; TSH, thyroid-stimulating hormone, VEGF: vascular endothelial growth factor. * The reference ranges of IL-6 and VEGF were <4.0 pg/mL and <38.3 pg/mL, respectively.

## Data Availability

All data are included in the main text.

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
