# Peer review of "Intracranial Hemorrhage in a Patient with TAFRO Syndrome Treated with Cyclosporine A and Rituximab"

_medicina, 2021, doi:10.3390/medicina57090971_

Round 1
Reviewer 1 Report
The manuscript presents in details an extraordinary case of the rare disorder TAFRO syndrome, which was firstly desribed in Japan 2010. It is focused on clinical and therapeutic , but not immune-pathophysiologic aspects.
The text might profit from the following:
- the list of references is highly focused on japanese experiences: check international publications and mention immunologic analyses/aspects (e.g. on B- and T-cells) and the role of platelets destruction in TAFRO and related disorders and add some international references (i.e. ref. Coutier et al. Ann Hematol 2018, p. 401f)
- Reduction of the amount of abbreviations would increase the fluent reading: omit TMA, iMCD, anti-SSA, anti.Jo-I, AKI.
- Minor points: Line 27: delete "suddenly", line 60: delete "to a nearly hospital", Table 1: add reference ranges to the mentioned cytokine levels, line 99: say the patient instead of "she", line 122 and 79: check VEGF levels (and reference ranges), which seems not concordant, line 110 and 112: use always the administered doses per kg b.w. as you do later on, line 124: delete "eventually", lines 162-165: unclear sentence together with the following sentence.
Author Response
Dear Reviewer #1
Thank you for your detailed reviewing on our manuscript. We are pleased that you presented very valuable comments and recommended publication with a minor revision. We have made every attempt to address your comments (please see point-by-point responses, below). Through addressing your comments, the manuscript is now much improved and acceptable to you for publication.
<Major points>
- The list of references is highly focused on Japanese experiences: check international publications and mention immunologic analyses/aspects (e.g. on B- and T-cells) and the role of platelets destruction in TAFRO and related disorders and add some international references (i.e. ref. Coutier et al. Ann Hematol 2018, p. 401f)
Response: We thank you for highlighting this important point. We agree to add the discussion related to recent international publications. We described the following paragraph in the Discussion section and added some international references: “Serum VEGF and circulating CD8+ T cells have been reported to be increased in non-Japanese patients with TAFRO syndrome [21-24]. Recent studies found that mTOR (mammalian target of rapamycin) signaling pathway, that was critical to VEGF expression and CD8+ T cell activation, was elevated in such patients [22,23]. Further, transcriptome analysis revealed signature genes associated with interferon type I (IFN-I), one of upstream regulators of mTOR signaling, were identified in CD8+ T cells, NK cells, and monocytes obtained from patients with TAFRO syndrome [24]. The overproduction of IFN-I in immune cells has been associated with the development of autoimmune diseases such as systemic lupus erythematous [25]. Notably, treatment with mTOR inhibitor has been shown to be effective in patients with tocilizumab-refractory TAFRO syndrome [22]. These findings suggest that IFN-I-induced activation of mTOR signaling pathway serves a central role in the pathogenesis of TAFRO syndrome and develops autoimmune features such as thrombocytopenia”.
- Reduction of the amount of abbreviations would increase the fluent reading: omit TMA, iMCD, anti-SSA, anti.Jo-I, AKI.
Response: As you pointed out, we returned “TMA, iMCD, anti-SSA, anti.Jo-I, and AKI” to “thrombotic microangiopathy, idiopathic multicentric Castleman’s disease, anti–Sjögren's-syndrome-related antigen A, and anti-histidyl-tRNA synthetase”, respectively.
<Minor points>
- line 27: delete "suddenly",
Response: We agree this point. We have deleted this word.
- line 60: delete "to a nearly hospital"
Response: We agree this point. We have deleted this word.
- table 1: add reference ranges to the mentioned cytokine levels
Response: As suggested, we have added the following sentence in table 1: “*The reference ranges of IL-6 and VEGF was <4.0 pg/mL and <38.3 pg/mL, respectively”.
- line 99: say the patient instead of "she"
Response: As you pointed out, we have reworded “the patient”.
- line 122 and 79: check VEGF levels (and reference ranges), which seems not concordant
Response: We agree that this sentence in line 122 was confusable. To make it more easily to understand, we have reworded the following sentence: “her serum VEGF level was still high at 92 pg/mL (reference range <38.3 pg/mL)”.
- line 110 and 112: use always the administered doses per kg b.w. as you do later on,
Response: As you pointed out, we have added “1 mg/kg of body weight” in these lines.
- line 124: delete "eventually”
Response: As suggested, we have deleted this word.
- lines 162-165: unclear sentence together with the following sentence.
Response: Thank you for your comments. We have reworded the following sentence: “An increasing number of case reports with TAFRO syndrome have recently described renal percutaneous biopsy among the diagnostic procedures, despite of low platelet counts”.
Reviewer 2 Report
Very interesting work!
Nothing to add.
Good for publication
Author Response
Dear Reviewer #2
Thank you very much for your reviewing our manuscript. We are pleased that you found this manuscript to be well performed and acceptable for publication.
